# Beta-2-Microglobulin Maintains Overall Survival Prediction in Binet A Stage Chronic Lymphocytic Leukemia Patients with Compromised Kidney Function in Both Treatment Eras of Chemoimmunotherapy and Targeted Agents

**DOI:** 10.3390/cancers16223744

**Published:** 2024-11-06

**Authors:** Jan-Paul Bohn, Valentina Stolzlechner, Georg Göbel, Wolfgang Willenbacher, Markus Pirklbauer, Normann Steiner, Dominik Wolf

**Affiliations:** 1Department of Internal Medicine V, Hematology and Oncology, Comprehensive Cancer Center Innsbruck (CCCI), Austrian Comprehensive Cancer Network (ACCN), Tyrolean Cancer Research Center (TKFI), Medical University of Innsbruck, 6020 Innsbruck, Austria; valentina.stolzlechner@student.i-med.ac.at (V.S.); wolfgang.willenbacher@i-med.ac.at (W.W.); normann.steiner@i-med.ac.at (N.S.); dominik.wolf@i-med.ac.at (D.W.); 2Department of Medical Statistics, Informatics and Health Economics, Medical University of Innsbruck, 6020 Innsbruck, Austria; georg.goebel@i-med.ac.at; 3Department of Internal Medicine IV, Nephrology and Hypertension, Medical University of Innsbruck, 6020 Innsbruck, Austria; markus.pirklbauer@i-med.ac.at

**Keywords:** chronic lymphocytic leukemia, CLL-IPI, prognosis

## Abstract

Precise and feasible prognostic scores are critical for design and enrollment of clinical trials in early-stage CLL patients at high-risk for inferior overall survival (OS). The CLL-IPI risk score is among the best validated models to predict OS in advanced-stage CLL patients receiving chemoimmunotherapy. However, data on its applicability in early-stage, Binet A, CLL patients as well as is in the modern treatment era of targeted agents are rare. Moreover, elevated beta-2-microglobulin (B2M) plasma levels account for two CLL-IPI scoring points, frequently upgrading patients to a higher risk group for inferior OS. Yet, B2M is commonly increased in patients with chronic kidney disease (CKD) per se and B2M-scoring in the CLL-IPI was not adjusted for CKD. Here we show that B2M plasma levels but not the CLL-IPI retain their prognostic value in Binet stage A CLL patients with compromised kidney function diagnosed in both treatment eras of chemoimmunotherapy and targeted agents.

## 1. Introduction

Chronic lymphocytic leukemia (CLL) is the most common leukemia in adults with an annual incidence rate of 6.4/100,000 [1]. Most patients present with asymptomatic, early-stage disease and are not in need of therapy [2]. However, the clinical course of CLL is very heterogeneous with some patients experiencing more aggressive disease [3]. Roughly one third of Binet A stage CLL patients develop symptomatic disease burden requiring specific treatment and may, ultimately, face inferior overall survival (OS) when compared to the general population [4,5]. Yet, the upfront identification of asymptomatic Binet A stage CLL patients at high risk for disease progression and inferior OS is very challenging [6,7,8,9,10]. The CLL international prognostic index (CLL-IPI) is currently the best validated model to predict OS in advanced-stage CLL patients receiving chemoimmunotherapy [11]. Increasing evidence suggests that it also allows to predict OS in asymptomatic early-stage CLL patients [5], although its predictive value of OS in CLL patients treated upfront with targeted agents remains less well defined [5,12,13,14]. Interestingly, increased beta-2-microglobulin (B2M) plasma levels account for two scoring points in the CLL-IPI and are given the same prognostic impact as an unmutated immunoglobulin heavy chain (*IGHV*) status—a well-established molecular feature associated with biologically more aggressive disease [15,16]. Thereby, patients are frequently allocated to a higher risk group for inferior survival in case of B2M plasma levels above 3.5 mg/L [17]. However, B2M plasma levels are commonly elevated in patients with chronic kidney disease (CKD) as a result of decreased glomerular filtration rate (GFR) [18] and the CLL-IPI risk model was not adjusted for compromised renal function [11,13].

Here, we report the long-term OS outcomes of 259 consecutive asymptomatic Binet A stage CLL patients from 2000 to 2022 and assess the prognostic utility of CLL-IPI risk stratification in patients with and without concurrent CKD. As OS in advanced-stage CLL patients has significantly improved with the introduction of novel agents into daily clinical practice, we next subdivided the observation period and compared OS outcomes in the treatment eras of chemoimmunotherapy (2000–2013) and targeted therapies (2014–2022). For the first time, we show that the CLL-IPI and elevated B2M plasma levels at the threshold of >3.5 mg/L retain their prognostic value in both CLL patients with and without concurrent CKD. Although the prognostic impact of the CLL-IPI on OS is lower in the era of targeted agents, our results demonstrate that elevated B2M plasma levels alone serve as an independent prognostic parameter in both treatment eras. Despite similar OS outcomes in the eras of chemoimmunotherapy and targeted agents, our data indicate that early-line treatment with target agents is associated with superior survival.

## 2. Materials and Methods

Newly diagnosed asymptomatic Binet A stage CLL patients not fulfilling iwCLL2018 criteria [4] for immediate treatment initiation were consecutively collected starting from January 2000 up until December 2022 at a tertiary care center in Innsbruck, Austria. Patients were identified from the clinical database of Innsbruck University Hospital and their OS outcomes were evaluated retrospectively. The study was approved by our institutional review board and by our ethics committee (protocol number 1206/2023) and was conducted in accordance with the Declaration of Helsinki in 1964 and its amendments. Two hundred and fifty-nine patients in total were given the diagnosis CLL based on WHO criteria by the presence of ≥5 × 10^9^/L clonal B lymphocytes with a typical CLL immunophenotype (i.e., CD5, CD19, CD20, and CD23) in the peripheral blood and Binet A stage disease defined as less than three palpably enlarged lymphoid sites without hemoglobin < 10 g/dL and platelets < 100 × 10^9^/L [19,20]. The CLL-IPI score was assessed for all patients for whom the required prognostic parameters were retrospectively evaluable at the time of diagnosis in accordance with the original publication of the German CLL study group [11]. The hierarchical multivariate risk model encompasses five prognostic variables with different assigned point scores: age > 65 years and Rai clinical stage I-IV yield one point each, B2M plasms levels > 3.5 mg/L and unmutated *IGHV* status yield two points each, and *TP53* disruption (*TP53* mutation of an allele frequency of ≥10% and/or deletion 17p) yields four points. The summed single ratings of these five factors equals the total individual CLL-IPI risk score (range, 0–10 points). This scoring system then allows the allocation of CLL patients into four major risk groups in terms of OS: low (scores 0–1), intermediate (scores 2–3), high (scores 4–6), and very high risk (scores 7–10). In our cohort, *IGHV* analysis was performed with next-generation sequencing (NGS) and used 98% germline identity as the cut-off to discriminate cases into mutated and unmutated *IGHV* [21,22]. Similarly, TP53 mutation analysis was conducted via targeted NGS with an allele frequency cut-off of ≥10% [23,24,25,26]. Patients were usually monitored for disease progression via physical examination and peripheral blood (PB) counts every 3 months in the first year after diagnosis, every 6 months for an additional 2 years, followed by annual visits in case of stable disease parameters [27]. Abdominal sonography was performed annually. Bone marrow evaluation was confined to cases with unclear cytopenia. During follow-up, CLL treatment was initiated according to iwCLL2018 [4] criteria, including progressive anemia and/or thrombocytopenia with cutoff levels of hemoglobin < 10 g/dL or platelet counts < 100 × 10^9^/L, symptomatic splenomegaly and/or lymphadenopathy, symptomatic extranodal disease involvement, recurrent infectious complications, lymphocyte doubling time < 6 months, and/or B-symptoms with or without significant fatigue (ECOG performance scale ≥ 2) [4]. Chronic kidney disease (CKD) is defined as abnormalities of kidney structure or function present for a minimum of 3 months. CKD is classified based on GFR category (G1–G5) and Albuminuria category (A1–A3) [28]. In our study, CKD was defined as an estimated GFR (eGFR) < 60 mL/min/1.73 m^2^ according to the Modification of Diet in Renal Disease (MDRD) equation [29], as this approach allowed us to correct for the effect of age, gender, race, and serum creatinine on GFR estimation. Moreover, it has been our local laboratory standard over the last decades and was readily available in all CLL patients analyzed. As the MDRD equation-based estimation of GFR was established to correctly estimate GFR below 60 mL/min/m^2^ (i.e., CKD stage 3 or higher), we only defined CLL patients with such moderate-to-severe impairment of kidney function as CKD for our analysis. Categorial variables were summarized as frequencies and percentages and compared using Fisher’s exact test or Chi-square test, depending on their distribution. The presence or absence of the normal distribution was assessed using the Kolmogorov–Smirnov test. Continuous variables were outlined as median values and range and compared using the Mann–Whitney U test. OS was measured from the date of CLL diagnosis until death of any cause. Observations of OS were censored at date of last contact for patients who were last known to be alive. OS estimates were determined by Kaplan–Meier analysis. OS for different subgroups was compared using the log-rank test. For the more detailed OS analysis, the observation period was subdivided into 2000–2013 (“Era of chemoimmunotherapy”) and 2014–2022 (“Era of targeted therapy”). The year of 2014 was chosen as the cut-off date due to the EMA approval and thus immediate availability and use of ibrutinib for CLL treatment in relapsed disease in Austria from 2014 onwards. Since May 2016, ibrutinib has been approved for frontline treatment. All analyses were performed using IBM SPSS Statistics 24 (New York, NY, USA) and GraphPad Prism 8 (San Diego, CA, USA).

## 3. Results

A total of 259 newly diagnosed CLL patients with Binet A stage disease not fulfilling iwCLL2018 [4] criteria for treatment initiation were identified. The median age at diagnosis was 66 years (range, 39–88 years) and the female-to-male ratio was 2:1. The incidence and distribution of genetic factors associated with early disease progression such as *TP53* disruptions (11.5%, n = 182) and unmutated *IGHV* mutational status (41.9%, n = 227) was similar to recently reported asymptomatic Binet A stage CLL patients within the CLL12 trial [30]. Our cohort included 16.9% (44/259) patients with CKD (eGFR < 60 mL/min). The median eGFR was 53 mL/min (range, 30–59 mL/min) and all 44 patients were categorized as CKD stage 3 (eGFR < 60 to 30 mL/min). These tended to be older than non-CKD patients (median age at diagnoses, 72 versus 64 years, *p* < 0.05) and had more often elevated B2M plasma levels > 3.5 mg/L (44.1% versus 10.6%, *p* < 0.001). Genetic high-risk parameters were overall well balanced between CKD and non-CKD patients (for more details on patient characteristics, see Table 1).

With a median follow-up of 97 months (range, 4–338 months), the median OS was 170 months (range, 146–194 months) for the whole study cohort. The median OS tended to be longer for female patients (218 months versus 158 months, *p* = 0.12) and generally decreased with increasing age (*p* < 0.001, Figure 1A). In CKD patients and non-CKD patients, the median OS was similar (*p* = 0.25, Figure 1B).

### 3.1. Overall Survival According to CLL-IPI Risk Groups and B2M Plasma Levels

Assessment of the CLL-IPI risk model was retrospectively feasible in 182/259 patients, of whom 45.6% were classified as low-risk, 31.3% as intermediate-risk, 16.5% as high-risk, and 6.6% as very high-risk for shorter OS. A total of 18.7% (34/182) of these patients presented with CKD.

For the whole observation period, the CLL-IPI facilitated prognostic segregation in terms of OS in both CKD (*p* = 0.02, Figure 2A) and non-CKD patients (*p* = 0.008, Figure 2B). In non-CKD patients (148/182), the median OS was 282 months for low-risk patients, 200 months (range, 138–262 months) for intermediate-risk patients, 170 months (range, 102–239 months) for high-risk patients, and 110 months (range, 65–155 months) for very high-risk patients. In CKD patients, the median OS was not reached for low-risk patients, 218 months for intermediate-risk patients, 81 months (range, 39–123 months) for high-risk patients, and 69 months (range, 67–70 months) for very high-risk patients.

As B2M plasma levels are commonly elevated in CKD patients [18] per se and elevated plasma levels account for two out of ten points in the weighted CLL-IPI risk score, we next analyzed the frequency and distribution of elevated B2M plasma levels in our cohort. The CLL-IPI recognizes increased B2M plasma levels with a threshold of >3.5 mg/L. Indeed, B2M plasma levels above this cut-off were documented significantly more often in CKD patients (44.1%, n = 15/34) than non-CKD patients (10.6%, n = 23/215). Nevertheless, B2M plasma level > 3.5 mg/L alone retained its prognostic impact on OS by effectively extracting two distinct subgroups also in CKD patients (*p* = 0.03, Figure 2C). Although B2M is more commonly elevated in CLL patients with concurrent CKD, these results further illustrate that the threshold of >3.5 mg/L B2M is indeed adequate to allow OS prediction in both CKD and non-CKD patients.

### 3.2. Overall Survival Comparison in the Era of Chemoimmunotherapy and Targeted Agents

OS prognosis in advanced-stage CLL patients has fundamentally improved since the introduction of novel chemotherapy-free targeted agents in the last decade, and OS analysis must be interpreted with caution along-side available treatment options for the patient cohort in question. Hence, we next investigated and compared the OS outcomes of our Binet A stage CLL patients in the era of chemoimmunotherapy (2000–2013) and targeted agents (2014–2022). All CLL patients diagnosed after 2013 and eventually meeting iwCLL2018 [2] criteria for specific treatment received novel agents either as first- and/or second-line therapy (for more details on treatment modalities see Table 1). Notably, OS was similar in both treatment periods (*p* = 0.87, Figure 3A). The CLL-IPI facilitated the separation of prognostic subgroups in CLL patients diagnosed before the year 2014 (*p* = 0.001, Figure 3B), whereas its impact on OS appears to diminish afterwards (*p* = 0.123, Figure 3C). Of interest, elevated B2M levels alone retained its prognostic value in the era of targeted therapy, including patients with concurrent CKD (*p* = 0.03, Figure 3D). Generally, the kidney function did not substantially change after the start of CLL treatment in our CKD patients in terms of altering CKD grading. Although OS outcomes did not significantly differ in both treatment eras overall, we further examined whether early-line treatment with targeted agents may be of benefit for OS. Indeed, OS was longest for patients having received novel agents frontline, whereas outcomes deteriorated with each additional previous treatment line given (*p* = 0.025, Figure 3E).

## 4. Discussion

Reviewing 23 years of clinical experience, we report here, for the first time, long-term OS outcomes in 259 consecutive Binet A stage patients in both eras of chemoimmunotherapy and targeted agents in the context of CLL-IPI risk stratification, elevated B2M plasma levels, and concurrent CKD. As expected, the CLL-IPI allowed for optimal prognostic separation in the era of chemoimmunotherapy, whereas its impact on OS prediction after introduction of targeted agents such as ibrutinib in 2014 clearly diminished. These results complement the recently reported reassessment of the CLL-IPI in a larger cohort of patients treated with novel agents demonstrating a preserved prognostic value in predicting progression-free survival but not in OS [13]. Surprisingly and despite being more commonly elevated in CKD patients, B2M retained its prognostic value not only in the treatment era of targeted agents, but also in CKD patients per se. As the CLL-IPI risk model was not adjusted for compromised kidney function in its original validation cohort [11], our data set provides the first evidence for its maintained applicability in predicting OS in CLL patients with concurrent CKD (eGFR < 60 mL/min/m^2^). Although the prognostic value of the CLL-IPI risk model on OS diminishes in patients treated upfront with novel agents, the retained prognostic impact of B2M on OS alone both in patients with CKD and in those treated with novel drugs strongly suggests the implementation of B2M as a promising covariate in up-coming CLL risk models predicting OS in the treatment era of novel drugs. The similar OS outcomes of our early-stage CLL patients in both treatment eras is in line with recently reported survival from the Danish National Chronic Lymphocytic Leukemia Register (DCLLR) in 2722 CLL patients with Binet A stage disease [12]. Yet, in the Danish cohort, reimbursement of targeted agents was confined to CLL patients with known *TP53* dysfunction in the treatment-naïve setting. In contrast, in our cohort, ibrutinib was reimbursed regardless of *TP53* status and line of treatment since May 2016. Hence, our data set may more accurately distinguish between both treatment eras in terms of broadly available novel agents. Although OS was favorable in both time periods, our results point to superior survival when targeted agents are given early in the course of symptomatic disease. Moreover, for reasons of data entry into the Danish register, the cut-off for B2M plasma levels was set at >4.0 mg/L, clearly above the threshold of 3.5 mg/L used within the CLL-IPI.

Our study has also concept-inherent limitations. First, the retrospective design itself restrains the power of our study. Second, the size of our study cohort is limited (n = 259) and only 44 patients (20%) presented with concurrent CKD. Yet, given a CKD prevalence of approximately 10% in the general population [31], the proportion of CLL patients with concurrent CKD appears rather high in our study. Additionally, we did not observe severe CKD grade 4 or higher (eGFR < 30 mL/min/m^2^) in our patients. Hence, we cannot exclude that B2M plasma levels retain their prognostic value for OS also in CLL patients with such severe CKD. Moreover, evaluation of the CLL-IPI risk score for OS was not universally feasible in all patients included in this analysis. However, even in 2024, the assessment of *IGHV* status and *TP53* aberrations at CLL diagnosis is still not recommended in the daily routine outside of clinical trials [4] and has not been broadly available at our institution before 2010. As such, the evaluable test results on *TP53* alterations and *IGHV* status in 70% of all 259 patients included in our real-world single-center study seems rather high. Furthermore, patient numbers of higher risk subgroups are small. Yet, the distribution of risk variables in our cohort is similarly balanced compared to the original risk model validation trial [11] and to recent prospective studies for early stage CLL patients, i.e., CLL12 [30]. Particularly, higher risk subgroups are generally small, with 29 patients in the “watch and wait” cohort of the original CLL-IPI validation study and 49 patients in the prospective CLL12 trial [11,30]. Hence, we still believe that our study cohort is of sufficient size to investigate the prognostic value of elevated B2M plasma levels in terms of OS in CKD and non-CKD Binet A stage CLL patients.

## 5. Conclusions

Our study demonstrates that elevated B2M plasma levels retain their prognostic value for OS in Binet A stage CLL patients with concurrent compromised kidney function diagnosed in both treatment eras of chemoimmunotherapy and novel agents. Precise and feasible prognostic scores are critical for design and enrollment of clinical trials in early-stage CLL patients at high-risk for inferior OS. As such, B2M still represents a promising covariate to be evaluated in up-coming prognostic models to identify patients at high risk for inferior OS in the era of targeted agents and regardless of kidney function.

## Figures and Tables

**Figure 1 cancers-16-03744-f001:**
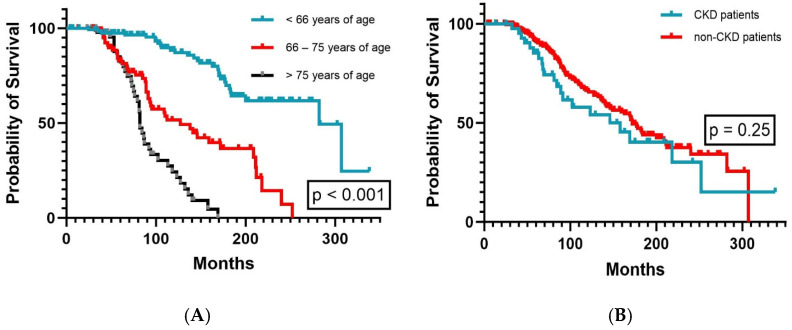
Overall survival according to (**A**) patient age (n = 259) and (**B**) patient kidney function (n = 259).

**Figure 2 cancers-16-03744-f002:**
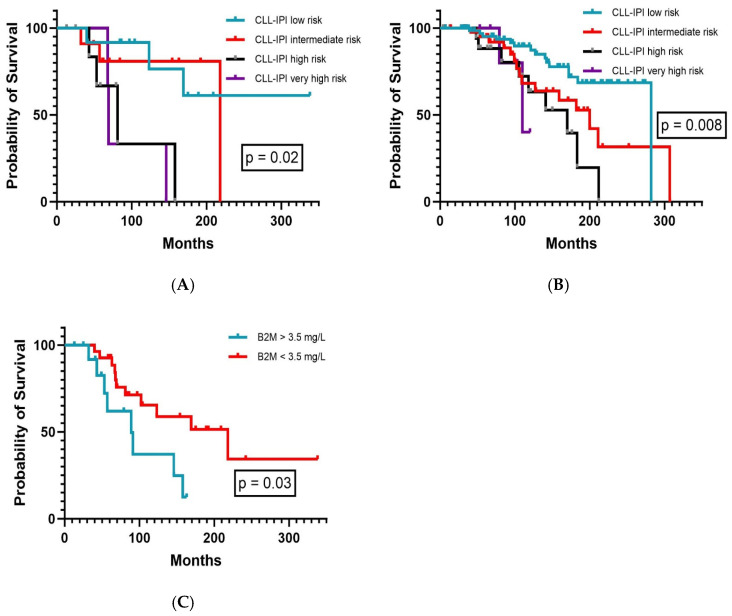
Overall survival according to (**A**) CLL-IPI in CKD patients (n = 34), (**B**) CLL-IPI in non-CKD patients (n = 148), and (**C**) beta-2-microglobulin plasma levels in CKD patients (n = 34).

**Figure 3 cancers-16-03744-f003:**
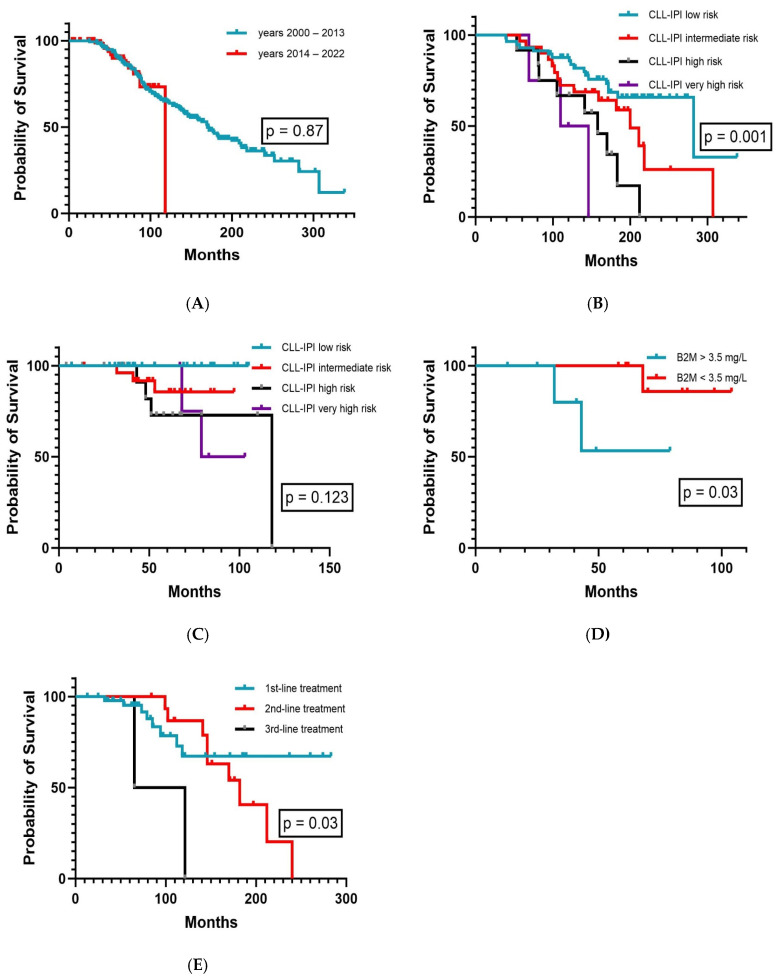
Overall survival according to (**A**) treatment eras (n = 259), (**B**) CLL-IPI in the era of chemoimmunotherapy (2000–2013, n = 104), (**C**) CLL-IPI in the era of targeted therapies (2014–2022, n = 78), (**D**) beta-2-microglobulin plasma levels in CKD patients in the era of targeted therapy (2014–2022, n = 16), and (**E**) line of treatment with targeted agents (n = 70).

**Table 1 cancers-16-03744-t001:** Patient characteristics (n = 259).

Parameter	Non-CKD Patients (n = 215)	CKD ^#^ Patients (n = 44)
Female	74 (34.1%)	18 (40.9%)
Median age (years)	64 (range, 39–88)	72 (range, 57–83)
Median eGFR * (mL/min/m^2^)	>60	53 (range, 30–59)
B2M ^+^ plasma level > 3.5 mg/L	23 (10.6%)	15 (44.1%)
*TP53* alterations	16 (10.8%, n = 148)	5 (14.7%, n = 34)
Unmutated *IGHV*	88 (40.5%, n = 148)	15 (44.1%, n = 34)
CLL-IPI score	148/215	34/44
Low risk	71 (48.0%)	12 (35.3%)
Intermediate risk	46 (31.1%)	11 (32.4%)
High risk	22 (14.9%)	8 (23.5%)
Very high risk	9 (6.1%)	3 (8.8%)
Start of CLL treatment in follow-up	108 (50.2%)	21 (47.7%)
1st/2nd line treatment with novel agents	58/108 (53.7%)	9/21 (42.9%)
Ibrutinib	38 (65.5%)	5 (55.5%)
Acalabrutinib	6 (10.3%)	2 (22.2%)
Zanubrutinib	4 (6.9%)	0
Venetoclax and Obinutuzumab	5 (8.6%)	2 (22.2%)
Ibrutinib and Venetoclax	5 (8.6%)	0

^#^ CKD, chronic kidney disease. * eGFR, MDRD equation-based estimation of glomerular filtration rate. ^+^ B2M, beta-2-microglobulin.

## Data Availability

The data presented in this study are available in this article.

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
