# Peer review of "Beta-2-Microglobulin Maintains Overall Survival Prediction in Binet A Stage Chronic Lymphocytic Leukemia Patients with Compromised Kidney Function in Both Treatment Eras of Chemoimmunotherapy and Targeted Agents"

_cancers, 2024, doi:10.3390/cancers16223744_

Round 1
Reviewer 1 Report
Comments and Suggestions for Authors
The authors analyzed the impact of B2M on the survival of early stage CLL patients including patients with CKD both in the era of chemoimmunotherapy and that of targeted agents, and found that B2M could be a valuable marker for the prognosis of these CLL patients regardless of CKD status. Although retrospective fashion, the results of this study are interesting, and I have only some minor comments.
Minor comments:
1. line 58: please define B2M with full term, then abbreviate thereafter throughout the manuscript.
2. line 64: glomerular filtration rate→ glomerular filtration rate (GFR) and same as above.
3. line 87: please don’t use Arabic numerals at the head of a sentence.
4. line 90: hemoglobin 100 g/dL→10 g/dL
5. line 90: platelets 100 G/l→100×109/L
6. line 96 : TP53→TP53 in all cases
7. line 112: glomerular filtration rate (GFR) category→GFR category
8. line 130: 2014→The year of 2014
9. legend for Table 1: Beta-2-Microglobulin→beta-2-microglobulin
Author Response
Response to Reviewer 1
The authors analyzed the impact of B2M on the survival of early stage CLL patients including patients with CKD both in the era of chemoimmunotherapy and that of targeted agents, and found that B2M could be a valuable marker for the prognosis of these CLL patients regardless of CKD status. Although retrospective fashion, the results of this study are interesting, and I have only some minor comments.
Minor comments:
- line 58: please define B2M with full term, then abbreviate thereafter throughout the manuscript.
- line 64: glomerular filtration rate→ glomerular filtration rate (GFR) and same as above.
Thank you for these important comments. Both B2M and GFR are now defined with full terms in lines 59 and 65, respectively.
- line 87: please don’t use Arabic numerals at the head of a sentence.
Thank you for this welcome addition. 259 has now been replaced by “Two hundred fifty-nine” in line 88.
- line 90: hemoglobin 100 g/dL→10 g/dL
- line 90: platelets 100 G/l→100×109/L
Thank you for these important corrections. Both labels have been adapted accordingly in lines 91-92.
- line 96 : TP53→TP53 in all cases
Thank you for this constructive remark. TP53 is now written in italic in all cases.
- line 112: glomerular filtration rate (GFR) category→GFR category
Thank you for this correction. The term is now abbreviated accordingly in line 117.
- line 130: 2014→The year of 2014
Thank you for this remark. The manuscript has been adapted accordingly in line 135.
- legend for Table 1: Beta-2-Microglobulin→beta-2-microglobulin
Thank you for this important correction. Table 1 has been adapted accordingly.
Reviewer 2 Report
Comments and Suggestions for Authors
The authors assess the interest of B2M in 259 patients with Binet stage A CLL and show that B2M > 3.5 mg/L is appropriate to predict OS in both CKD and non-CKD patients.This remains true in the era of targeted treatments.
For improving the manuscript, a few comments/suggestions
Major points.
We do not know the type of the CKD. Did you perform renal biopsies and what is the real nature of diseases? You did not observe severe renal insufficiency and the eGFR was between 30 and 60 in all your patients. Can you eliminate selection bias?
The study is retrospective and CLL-IPI risk was evalutaed in only 182/259 patients (70%). Can you explain why? Are the patients you studied are consecutive or not? You compare the frequency of IGVH and TP53 with a clinical trial. Is your study a real life study or not?
The number of patients with CKD is low and it is also a limit of the study. You have to highlight this point in the summary.
Like your study is retrospective, is it possible to obtain the results of renal fonction after treatment. Are they better or not? This can be added value for the article even if the study design was not made for this.
Minor points
In the discussion, please shorten. The first paragraph is redundant.
Why did you choose MDRD and not CDK-EPI. Explain your choice?.
Author Response
Response to Reviewer 2
The authors assess the interest of B2M in 259 patients with Binet stage A CLL and show that B2M > 3.5 mg/L is appropriate to predict OS in both CKD and non-CKD patients.This remains true in the era of targeted treatments.
For improving the manuscript, a few comments/suggestions
Major points.
1) We do not know the type of the CKD. Did you perform renal biopsies and what is the real nature of diseases?
Thank you for this important comment. Based on the retrospective character of our analysis and the fact that CKD cause was not routinely assessed during regular hematology visits we cannot provide detailed information on the underlying biology of CKD in our study population. However, with respect to elevated B2M plasma levels our analysis primarily focuses on impaired renal function, i.e. reduced eGFR, which is irrespective of its actual underlying cause.
2) You did not observe severe renal insufficiency and the eGFR was between 30 and 60 in all your patients. Can you eliminate selection bias?
Thank you for this meaningful remark. Indeed, we did not observe severe CKD in terms of grade 4 or higher (eGFR < 30 ml/min). We now commented on this limitation in the discussion chapter in lines 250-253. However, there is no selection bias as all patients included in our analysis presented consecutively at our center. No asymptomatic Binet A stage CLL patient at our center was excluded from the analysis.
3) The study is retrospective and CLL-IPI risk was evalutaed in only 182/259 patients (70%). Can you explain why? Are the patients you studied are consecutive or not?
Thank you for this insightful comment. As testing for TP53 dysfunction and IGHV status at CLL diagnosis is not generally recommend outside of clinical trials, we did not have test results evaluable in all patients included in this retrospective analysis. As such, we could not calculate the CLL-IPI in patients with unknown status of TP53 and IGHV status. All patients included in this analysis were consecutive. However, it was open to the treating physician’ discretion to evaluate these genetic tests in individual patients. Moreover, genetic testing for TP53 mutations and IGHV status was not broadly available at our institution before 2010. As the analysis reflects on the clinical outcomes of CLL patients diagnosed from 2000 – 2022, this also explains why we cannot provide these data for all patients. We now added this important information as limitation in the discussion chapter in lines 253 – 258.
4) “The number of patients with CKD is low and it is also a limit of the study. You have to highlight this point in the summary.
Thank you for this important remark. Indeed, only 44 patients (20%) presented with concurrent CKD in our study. However, given a CKD prevalence of about 10% in the general population, the proportion of CKD patients included in our study appears rather high. This meaningful addition is now discussed and cited in lines 248 – 250.
5) “Like your study is retrospective, is it possible to obtain the results of renal fonction after treatment. Are they better or not? This can be added value for the article even if the study design was not made for this.”
Thank you for this meaningful comment. We now re-checked the eGFR values in all patients after start of treatment. Here, we did not observe major changes in kidney function in terms of altering CKD grading. We added this very useful information in lines 203-205.
Minor points
6) “In the discussion, please shorten. The first paragraph is redundant.”
Thank you for this important addition. The first paragraph of the discussion has now been deleted.
7) “Why did you choose MDRD and not CDK-EPI. Explain your choice?”
Thank you for the insightful comment. In our study, CKD was defined as an estimated GFR (eGFR) < 60 ml/min/1.73 m² according to the Modification of Diet in Renal Disease (MDRD) equation as this approach allowed us to correct for the effect of age, gender, race, and serum creatinine on GFR estimation. Moreover, it has been our local laboratory standard over the last decades and was readily available in all CLL patients analyzed. Of course, CDK-EPI would also be a very good option, but it was not readily available at our institution’s laboratory. We commented on our choice in the Methods section, lines 118 – 122.
Reviewer 3 Report
Comments and Suggestions for Authors
Beta-2-Microglobulin maintains overall survival prediction in Binet A stage CLL patients with compromised kidney function in both treatment eras of chemoimmunotherapy and targeted agents.
Jan-Paul Bohn and colleagues.
Thank you for asking me to review the above MS. I must apologize for the slight delay in my reply.
Beta-2 microglobulin (B2M) is an important component of prognostic schemes in chronic lymphocytic leukemia (CLL). The influence of raised B2M due to renal function impairment on prognostic evaluation in CLL patients is unclear. Here, Bohn and colleagues have evaluated a cohort of 259 patients with clinical stage A CLL not requiring immediate treatment seen at a single center in Innsbruck, Austria between January 2000 and December 2022. Their principal conclusion is that B2M retains prognostic significance in the face of renal impairment, even for patients receiving precision medicines, which is of some interest.
The MS is well written and the data are well presented.
The major limitation of this study (acknowledged by the authors) is the relatively small sample size – this is a single center study. Compare this with the 2722 patients recorded in the Danish national study (doi:10.1182/bloodadvances.2021006259.). Only 15 patients with CKD had a raised B2M. Another possible limitation is that 50% of the patients were started on therapy during relatively short period of FU; this seems to me to be quite high.
Minor points
How and where were IGHV analysis and interpretation performed please?
TP53 and IGHV should be in italics please
What was the rationale for performing “abdominal sonography” (??Ultrasound) every year please? (line 104)
Author Response
Response to Reviewer 3
Beta-2 microglobulin (B2M) is an important component of prognostic schemes in chronic lymphocytic leukemia (CLL). The influence of raised B2M due to renal function impairment on prognostic evaluation in CLL patients is unclear. Here, Bohn and colleagues have evaluated a cohort of 259 patients with clinical stage A CLL not requiring immediate treatment seen at a single center in Innsbruck, Austria between January 2000 and December 2022. Their principal conclusion is that B2M retains prognostic significance in the face of renal impairment, even for patients receiving precision medicines, which is of some interest.
The MS is well written and the data are well presented.
1) The major limitation of this study (acknowledged by the authors) is the relatively small sample size – this is a single center study. Compare this with the 2722 patients recorded in the Danish national study (doi:10.1182/bloodadvances.2021006259.). Only 15 patients with CKD had a raised B2M.
Thank you for this important comment. Indeed, our patient sample size as a single-center study is clearly smaller than the Danish national study. However, and as mentioned in the discussion section, reimbursement of novel agents since 2014 has been far less restricted in Austria compared to Denmark, a clear advantage in the context of overall survival comparison in the era of targeted drugs. Additionally, the cut-off for B2M in the Danish national study was different to that being used within the CLL-IPI. This issue is commented on in detail in the discussion section.
2) “Another possible limitation is that 50% of the patients were started on therapy during relatively short period of FU; this seems to me to be quite high.”
Thank you for this meaningful remark. Indeed, we report treatment start in about 50% of patients during a median follow-up of 97 months. However, in the recent CLL12-trial in Binet A stage CLL patients, 3-year event-free survival was only 60.1% in the placebo group (doi.org/10.1182/blood.2021010845). Taken these prospective data into account, start of treatment in 50% of patients after median follow-up of nearly 7.5 years does not appear too high after all.
Minor points
3) “How and where were IGHV analysis and interpretation performed please?”
Thank you for this important addition. We performed IGHV analysis with next-generation sequencing using 98% germline identity as cut-off to discriminate between mutated and unmutated cases. This valuable information has now been added and cited in lines 102 – 104.
4) “TP53 and IGHV should be in italics please”
Thank you for the valuable comment. We adapted the manuscript accordingly.
5) “What was the rationale for performing “abdominal sonography” (??Ultrasound) every year please? (line 104)”
Thank you for this critical comment. At our institution, abdominal ultrasound is performed annually to more objectively document dynamics in size of spleen or lymph nodes in all CLL patients than would possible with physical examination alone. But this is not generally recommend in current CLL management guidelines.
Round 2
Reviewer 2 Report
Comments and Suggestions for Authors
No additionnal comment and many thanks to the authors to modify the manuscript according to the suggestions of the reviewer.